# Diffusing Differentiable Representations

**Yash Savani**
Carnegie Mellon University
ysavani@andrew.cmu.edu

**Marc Finzi**
Carnegie Mellon University
mfinzi@andrew.cmu.edu

**J. Zico Kolter**
Carnegie Mellon University
zkolter@andrew.cmu.edu

## Abstract

We introduce a novel, training-free method for sampling *differentiable representations* (diffreps) using pretrained diffusion models. Rather than merely mode-seeking, our method achieves sampling by "pulling back" the dynamics of the reverse-time process—from the image space to the diffrep parameter space—and updating the parameters according to this pulled-back process. We identify an implicit constraint on the samples induced by the diffrep and demonstrate that addressing this constraint significantly improves the consistency and detail of the generated objects. Our method yields diffreps with substantially **improved quality and diversity** for images, panoramas, and 3D NeRFs compared to existing techniques. Our approach is a general-purpose method for sampling diffreps, expanding the scope of problems that diffusion models can tackle.

## 1 Introduction

Diffusion models have emerged as a powerful tool for generative modeling [Ho et al., 2020, Song et al., 2022, 2021, Karras et al., 2022, Rombach et al., 2022], and subsequent work has extended these models to generate complex objects—such as 3D assets, tiled images, and more. Although such approaches typically require training or at least fine-tuning of the diffusion models for these new modalities [Wang et al., 2023, Luo et al., 2023], two notable exceptions, Wang et al. [2022] and Poole et al. [2022], have developed methods for *training-free* production of 3D objects by *directly* using an image-based diffusion model. Both methods work by optimizing a *differentiable representation* (diffrep)—in this case, a Neural Radiance Field (NeRF) [Mildenhall et al., 2020]—to produce rendered views consistent with the output of the image-based diffusion model. Unfortunately, the nature of both these methods is that they optimize the diffrep to produce the "most likely" representation consistent with the images; that is, they perform *mode-seeking* rather than actually *sampling* from the diffusion model. This results in overly smoothed outputs that lack detail and do not reflect the underlying distribution of the diffusion model.

In this paper, we present a novel method for sampling *directly* in the diffrep space using pretrained diffusion models. The method is training-free, can handle arbitrary diffreps, and performs *true sampling* according to the underlying diffusion model rather than merely mode-seeking. The key idea of our approach is to rewrite the reverse diffusion process itself in the diffrep parameter space. This is achieved by "pulling back" the dynamics of the reverse time process—from the image space to the parameter space—and solving a (small) optimization problem, implied by the pulled-back dynamics, over the parameters for each diffusion step. To further encourage solver convergence, we identify constraints that the diffrep induces on the samples and use these constraints to guide the reverse process to generate samples from the high-density regions of the diffusion model while satisfying the constraints along the sampling trajectory.

Our experiments use a pretrained image-based diffusion model (Stable Diffusion 1.5 [Rombach et al., 2022]) to generate samples from various classes of diffreps. For example, by sampling SIREN representations [Vincent, 2011] with wrap-around boundary constraints, we can sample total 360-degree panorama scenes, and by sampling NeRF representations [Mildenhall et al., 2020], we can

generate 3D models of many objects. In both settings—as well as in baseline comparisons on simple SIREN-based (non-panorama) image generation—our approach substantially outperforms previous methods such as Score Jacobian Chaining (SJC) [Wang et al., 2022]. Though the problem setting is considerably more difficult without fine-tuning or retraining, sampling diffreps using pretrained diffusion models with our method substantially improves and extends the state-of-the-art in training-free generation methods.

## 2 Related work and preliminaries

### 2.1 Differentiable representations (diffreps)

Diffreps are a powerful tool for representing complex scenes or manifolds using parameterized differentiable functions to map coordinates of the scene into a feature space that encodes the properties of the scene at those coordinates. Popular instantiations of diffreps include: **SIRENs** [Sitzmann et al., 2020], which implicitly model an image using an MLP with sinusoidal activations to map 2D $(x, y)$ pixel coordinates into RGB colors, and **NeRFs** [Mildenhall et al., 2020], which implicitly model a 3D scene using an MLP that transforms 3D $(x, y, z)$ voxel coordinates with view directions $(\phi, \psi)$ into $RGB\sigma$ values. We can render images from different views of the scene by numerically integrating the NeRF outputs along the unprojected rays from the camera.

Faster or alternative diffreps also exist for 3D scenes—such as the InstantNGP [Müller et al., 2022] model or Gaussian splats [Kerbl et al., 2023]—although in this work we focus on the basic NeRF architecture. Many other kinds of visual assets can also be formulated as diffreps. For example, we can use diffreps to model panoramas, spherical images, texture maps for 3D meshes, compositions of multiple images, scenes from a movie, and even the output of kinematic and fluid simulations.

Many interesting diffreps can be used to render not just one image from the scene but *multiple coupled images* or even a *distribution over images*. Given a diffrep, parameterized by $\theta \in \Theta$, for a scene—such as a SIREN panorama or a NeRF—we can render an image of the scene from a view $\pi \in \Pi$ using a differentiable render function $f(\theta, \pi) = \texttt{image} \in \mathcal{X}$. To accommodate the multiview setting in our discussion, we consider the "curried" Haskell form of the render function $f : \Theta \to (\Pi \to \mathcal{X})$, where $f(\theta) : \Pi \to \mathcal{X}$ is a map itself from a view $\pi \in \Pi$ to $f(\theta)(\pi) = f(\theta, \pi) = \texttt{image} \in \mathcal{X}$.

In the case of NeRFs or SIREN panoramas, the view $\pi$ is a continuous random variable drawn from a distribution that, with abuse of notation, we will also call $\Pi$. To formalize this, let $\mathcal{H} \subseteq \mathcal{X}^\Pi$ be a vector space of functions from $\Pi$ to $\mathcal{X}$. With this definition, we can write the signature of $f$ as $f : \Theta \to \mathcal{H}$. Although $\mathcal{H}$ is usually larger than $\mathcal{X}$ (and can be even infinite in some cases), we can simplify our notation by equipping $\mathcal{H}$ with an inner product to make it a Hilbert space. This allows us to use familiar matrix notation with $\mathcal{H}$ and hide the view dependence of $f(\theta)$ in $\mathcal{H}$.

To complete this formalization, we lift the inner product on $\mathcal{X}$ to define an inner product on $\mathcal{H}$:

$$\forall h, g \in \mathcal{H} : \quad \langle h, g \rangle := \mathbb{E}_{\pi \sim \Pi}[\langle h(\pi), g(\pi) \rangle] = \mathbb{E}_{\pi \sim \Pi}[h(\pi)^\top g(\pi)].$$

This formulation allows us to handle the multiview nature of NeRFs and SIREN panoramas in a mathematically rigorous way while preserving the convenient notation.

The partial application of $f$ returns an entire view-dependent function $f(\theta) \in \mathcal{H}$. Given a set of images $(x_i)_{i \in [N]}$ of the scene from different views $(\pi_i)_{i \in [N]}$, the standard diffrep task is to find a $\theta \in \Theta$ that minimizes $\mathcal{L}(\theta) = \sum_{i \in [N]} \|f(\theta)(\pi_i) - x_i\|$. Because the diffreps and $f$ are both differentiable, this can be accomplished using first-order solvers, such as gradient descent or Adam.

For pedagogical convenience, we identify $\mathcal{H}$ with $\mathcal{X}$ in the following sections. This identification simplifies the presentation, provides a more interpretable and intuitive perspective on our method, and is precise when $\Pi$ is a singleton. Because $\mathcal{H}$ is a Hilbert space, the main points of our arguments still hold when we consider a larger $\Pi$. We describe the specific changes needed to adapt our method to the general multiview setting when $\mathcal{H} \neq \mathcal{X}$, in subsection 3.2.

### 2.2 Diffusion models

Diffusion models implicitly define a probability distribution via "reversing" a forward noising process. While many different presentations of image diffusion models exist in the literature (e.g., DDPM

[Ho et al., 2020], DDIM [Song et al., 2022], Score-Based Generative Modeling through SDEs [Song et al., 2021], EDM [Karras et al., 2022]), they are all equivalent for our intended purposes.

Given a noise schedule $\sigma(t)$ for $t \in [0, T]$ and a score function $\nabla \log p_t(x(t))$ for the distribution $p_t$ over noisy images $x(t)$ at time $t$, we can sample from $p_0$ by first initializing $x(T) \sim \mathcal{N}(0, \sigma^2(T)I)$ and then following the reverse time probability flow ODE (PF-ODE) given in Karras et al. [2022] to transform the easy-to-sample $x(T)$ into a sample from $p_0$:

$$\frac{dx}{dt} = -\dot{\sigma}(t)\sigma(t)\nabla \log p_t(x(t)). \tag{1}$$

The PF-ODE is constructed so that the perturbation kernel (conditional distribution) is given by $p_{0t}(x(t)|x(0)) = \mathcal{N}(x(t); x(0), \sigma^2(t)I)$. Using the reparameterization trick, we can write this as $x(t) = x(0) + \sigma(t)\epsilon$, where $\epsilon \sim \mathcal{N}(0, I)$.

We approximate the score function with a (learned) noise predictor $\hat{\epsilon}(x(t), t) \approx \frac{x(t) - \mathbb{E}[x(0)|x(t)]}{\sigma(t)}$ using the Tweedie formula [Robbins, 1956, Efron, 2011]: $\nabla \log p_t(x(t)) = \frac{\mathbb{E}[x(0)|x(t)] - x(t)}{\sigma^2(t)} \approx -\frac{\hat{\epsilon}(x(t), t)}{\sigma(t)}$.

### 2.3 Training differentiable representations using diffusion priors

#### 2.3.1 Training-free methods

Poole et al. [2022] laid the foundation, demonstrating that generating 3D assets from purely 2D image-trained diffusion models was *even possible*. In **DreamFusion (SDS)**, they perform gradient ascent using $\mathbb{E}_{t,\epsilon}[w(t)J^\top(\hat{\epsilon}(f(\theta) + \sigma(t)\epsilon, t) - \epsilon)]$, derived from the denoising objective, where $J$ is the Jacobian of the differentiable render function $f$ and $\hat{\epsilon}$ is the learned noise predictor.

Independently, Wang et al. [2022] introduced **Score Jacobian Chaining (SJC)**, which performs gradient ascent using $\nabla \log p(\theta) := \mathbb{E}_{t,\epsilon}[J^\top \nabla \log p_t(f(\theta) + \sigma(t)\epsilon)]$ with a custom sampling schedule for the $t$s. The custom schedule can be interpreted as gradient annealing to align the implicit $\sigma(t)$ of the diffrep with the $t$ used to evaluate the score function.

**Comparison of methods**  Using the Tweedie formula, we can rewrite the SDS objective as $\mathbb{E}_{t,\epsilon}\left[\frac{w(t)}{\sigma(t)}J^\top \nabla \log p_t(f(\theta) + \sigma(t)\epsilon)\right]$[1]. This expression is identical to the $\nabla \log p(\theta)$ term from SJC if we let $w(t) = \sigma(t)$ for all $t$. Both methods follow this gradient to convergence, which leads to a critical point—a local maximum or *mode*—in the $\log p(\theta)$ landscape. To approximate the objective, both approaches use Monte Carlo sampling.

Both approaches optimize the objective using gradient ascent (GA). While GA on the transformed score resembles solving the PF-ODE–particularly in their discretized forms–the two procedures serve fundamentally different purposes. GA cares only about finding a local maximum in the $\log p(\theta)$ landscape, regardless of the trajectory taken to get there. In contrast, the PF-ODE follows a specific path to ensure that generated samples are typical of the distribution. For a discussion of the limitations of GA and why it does not produce representative samples from the distribution, see Appendix A.

**Recent developments**  Several subsequent works have built upon the SDS and SJC methods by incorporating additional inputs, fine-tuning, and regularization to enhance the quality of the generated 3D assets. Zero-1-to-3 [Liu et al.] expands on SJC by fine-tuning the score function to leverage a single real input image and additional view information. Magic 123 [Qian et al.] further builds on this by incorporating additional priors derived from 3D data. Fantasia3D [Chen et al.] separates geometry modeling and appearance into distinct components, using SDS to update both. HiFA [Zhu et al.] introduces a modified schedule and applies regularization to improve the SDS process. Finally, LatentNeRF [Metzer et al.] utilizes SDS but parameterizes the object in the latent space of a stable diffusion autoencoder, rendering into latent dimensions rather than RGB values. While these methods rely on the SDS/SJC framework for mode-seeking, our work takes a wholly different approach, focusing on developing a more faithful sampling procedure to replace SDS/SJC for both 3D generation and broader differentiable function sampling.

---

[1]Note that $\mathbb{E}[\epsilon] = 0$, so the additional $-\epsilon$ term in the SDS objective behaves as an unbiased variance reduction term for the score estimate since $\text{Cov}(\hat{\epsilon}, \epsilon) > 0$.

### 2.3.2 Pretrained and fine-tuning methods

Unlike SDS and SJC—which are entirely zero-shot and can be performed with a frozen diffusion model—several other methods have been developed to achieve improved quality and diversity of the generated diffreps, albeit at the cost of additional fine-tuning. While some of these methods require additional data [Zhang et al.], ProlificDreamer (VSD) [Wang et al., 2023] and DiffInstruct [Luo et al., 2023] are two examples in which diffusion models can be fine-tuned or distilled using only synthetically generated data.

VSD specifically addresses the problem of generating 3D assets from a 2D diffusion model using particle-based variational inference to follow the Wasserstein gradient flow, solving the KL objective from SDS in the $\mathbb{W}_2(\Theta)$ parameter distribution space. This approach produces high-quality, diverse samples even at lower guidance levels.

In our work, we restrict ourselves to the *training-free setting*. This choice more readily enables application to new modalities and is independent of the score function estimator's architecture.

### 2.4 Constrained sampling methods

Our method requires generating multiple consistent images of a scene from different views. For instance, we generate various images of a 3D scene from different camera locations and orientations to determine NeRF parameters. These images must be consistent to ensure coherent updates to the diffrep parameters. While the diffrep inherently enforces this consistency, we can improve our method's convergence rate by encouraging the reverse process to satisfy consistency conditions using constrained sampling methods.

Several approaches enable constrained sampling from an unconditionally trained diffusion model. The naïve approach of projecting onto constraints [Song et al., 2021] leads to samples lacking global coherence. In contrast, Lugmayr et al. [2022] propose RePaint—a simple method for inpainting that intermingles forward steps with reverse steps to harmonize generated noisy samples with constraint information. For references to more general methods used to constrain the distribution by conditioning the score function, see Appendix B.

## 3 Pulling back diffusion models to sample differentiable representations

The score model associated with the noise predictor implicitly defines a distribution on the data space $\mathcal{X}$, and the reverse time PF-ODE from Eq. 1 provides a means to sample from that distribution. In this section, we will show how to use the score model and the PF-ODE to sample the parameters of a differentiable representation (diffrep) so that the rendered views resemble samples from the implicit distribution.

SDS and SJC derive their parameter updates using the pullback of the sample score function through the render map $f$. This pullback is obtained using the chain rule: $\frac{d(\log p)}{d\theta} = \frac{d(\log p)}{dx}\frac{dx}{d\theta}$. The pullback score is then used *as is* for gradient ascent in the parameter space. However, since $p$ is a distribution—not a simple scalar field—the change of variables formula requires an additional $\log \det J$ volume adjustment term for the correct pulled-back score function in the parameter space, where $J$ is the Jacobian of $f$. A careful examination of this approach through the lens of differential geometry reveals even deeper issues.

In differential geometry, it is a vector field (a section of the tangent bundle $T\mathcal{X}$)—not a differential 1-form (a section of the cotangent bundle $T^*\mathcal{X}$)—that defines the integral curves (flows) on a manifold. To derive the probability flow ODE in the parameter space, we must pull back the sample vector field $\frac{dx}{dt} \in T\mathcal{X}$ to the parameter vector field $\frac{d\theta}{dt} \in T\Theta$ using $f$. The pullback of the vector field through $f$ is given by $(f^* \frac{dx}{dt})|_\theta = (J^\top J)^{-1} J^\top \frac{dx}{dt}|_{f(\theta)}$, Thus, as we derive in Appendix C, the pullback of the probability flow ODE is:

$$\frac{d\theta}{dt} = -\dot{\sigma}(t)\sigma(t)(J^\top J)^{-1} J^\top \nabla \log p_t(f(\theta)). \tag{2}$$

We note that this is in contrast to the score chaining from Poole et al. [2022], Wang et al. [2022]. The confusion stems from comparing the different types of pulled-back elements. While $\frac{dx}{dt}$ in Eq. 1

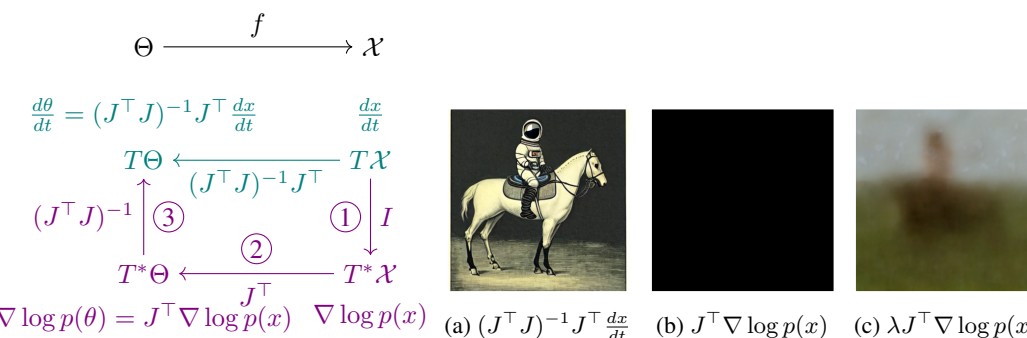

$$\Theta \xrightarrow{\quad f \quad} \mathcal{X}$$

$$\frac{d\theta}{dt} = (J^\top J)^{-1} J^\top \frac{dx}{dt} \qquad\qquad \frac{dx}{dt}$$

$$T\Theta \xleftarrow{\quad (J^\top J)^{-1} J^\top \quad} T\mathcal{X}$$

$$(J^\top J)^{-1} \bigg\uparrow \;③ \qquad\qquad ① \bigg\downarrow I$$

$$T^*\Theta \xleftarrow{\quad ② \quad}_{J^\top} T^*\mathcal{X}$$

$$\nabla \log p(\theta) = J^\top \nabla \log p(x) \qquad \nabla \log p(x)$$

(a) $(J^\top J)^{-1} J^\top \frac{dx}{dt}$   (b) $J^\top \nabla \log p(x)$   (c) $\lambda J^\top \nabla \log p(x)$

Figure 1: **(Left)** Commutative diagram showing how the PF-ODE vector field gets pulled back through $f$, respecting the differential geometry. The process involves: ① converting $\frac{dx}{dt}$ to the cotangent vector field $\nabla \log p(x)$ (up to scaling terms) with the Euclidean metric $I$, ② pulling back $\nabla \log p(x)$ via the chain rule using the Jacobian $J$, and then ③ transforming the pulled back differential form score function into the corresponding vector field using the inverse of pulled back metric $(J^\top J)^{-1}$. When used in a PF-ODE, SJC and SDS take the bottom path with the chain rule, however they do not complete the path by neglecting the $T^*\Theta \to T\Theta$ transformation. **(Right)** SIREN image renders generated using the PF-ODE schedule with the prompt "An astronaut riding a horse" using the: (a) complete pulled-back $\frac{dx}{dt}$ vector field, (b) pulled-back covector field from SJC (omitting step ③) $J^\top \nabla \log p(x)$, (c) Scaled pulled-back covector field from SJC $\lambda = 0.0001$.

is a vector field, the score function $\nabla \log p_t(x(t))$ is a covector field—a differential form. Pulling back the score function as a differential form correctly yields $J^\top \nabla \log p_t(f(\theta))$—the term used in SJC. The issue lies with the hidden (inverse) Euclidean metric within Eq. 1, which converts the differential-form score function into the corresponding vector field.

In canonical coordinates, the Euclidean metric is the identity. Consequently, the components of the score function remain unchanged when transformed into a vector field. Therefore we can safely ignore the metric term in the PF-ODE formulation of Eq. 1 for the sample space $\mathcal{X}$. However, this is not true for the diffrep parameter space $\Theta$.

To convert the pulled-back score function into the corresponding pulled-back vector field, we must use the pulled-back inverse Euclidean metric given by $(J^\top J)^{-1}$. This yields the pulled-back form of the PF-ODE in Eq. 2[2]. Fig. 1 illustrates this procedure via a commutative diagram and provides an example of what goes wrong if you use the incorrect pulled-back term, as suggested by SDS and SJC. In the SDS and SJC approaches, the $(J^\top J)^{-1}$ term behaves like a PSD preconditioner for gradient ascent. It may accelerate the convergence rate to the mode but does not fundamentally change the solution. However, this term is critical for the pulled-back PF-ODE since it impacts the entire trajectory and the ultimate sample (see Appendix A for more discussion). For an explanation of why the $\log \det J$ term is absent in the mathematically correct update, see Appendix C.

A more intuitive way to understand why the correct update is given by Eq. 2 rather than SDS and SJC's version is to consider the case where the input and output dimensions are equal. From the chain rule, we have $\frac{d\theta}{dt} = \frac{d\theta}{dx}\frac{dx}{dt}$. Using the inverse function theorem, we compute $\frac{d\theta}{dx} = \left(\frac{dx}{d\theta}\right)^{-1} = J^{-1}$, where $J$ is $f$'s Jacobian. This leads to $\frac{d\theta}{dt} = J^{-1}\frac{dx}{dt}$. For invertible $f$, this equation and Eq. 2 are equivalent. For non-invertible $f$, we can interpret the pullback as the solution to the least-squares minimization problem $\arg\min_{\frac{d\theta}{dt}} \left\| J\frac{d\theta}{dt} - \frac{dx}{dt} \right\|^2$.

## 3.1 Efficient implementation

**Separated noise**   Following the pulled-back PF-ODE in Eq. 2, we can find $\theta_t$ such that $f(\theta_t)$ represents a sample $x(t) \sim p_{0t}(x(t)|x(0)) = \mathcal{N}(x(t); x(0), \sigma^2(t)I)$. However, this approach has a limitation: $x(t)$ is noisy, and most diffrep architectures are optimized for typical, noise-free images. Consequently, $J$ is likely ill-conditioned, leading to longer convergence times.

---

[2]One may be tempted to impose that the metric is Euclidean both in the $\mathcal{X}$ space and the $\Theta$ space, but these two choices are incompatible because the render transformation maps between the two spaces.

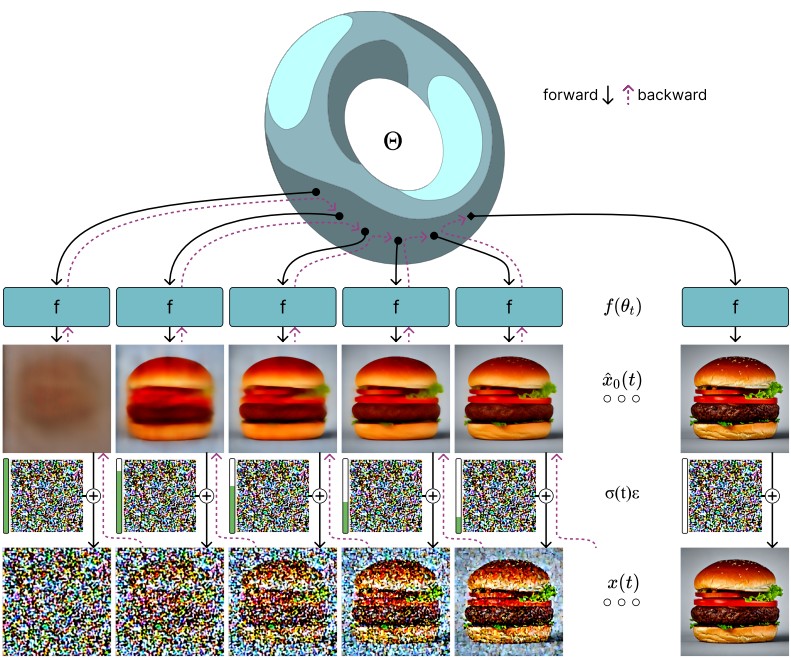

Figure 2: The parameters of the diffrep $\theta_t \in \Theta$ (torus) are used to render the noiseless signal $f(\theta_t) = \widehat{x}_0(t)$, which are then combined with the noise $\sigma(t)\epsilon$ to generate the noisy sample $x(t)$. We can pull back each step of the reverse diffusion process to update the parameters $\theta_t + \Delta\theta_t$.

To address this issue, we factor $x(t)$ into a noiseless signal and noise using the reparameterization of the perturbation kernel: $x(t) = \widehat{x}_0(t) + \sigma(t)\epsilon(t)$. By letting $\epsilon(t) = \epsilon$ remain constant throughout the sampling trajectory and starting with $\widehat{x}_0(T) = 0$, we can update $\widehat{x}_0$ with $\frac{d\widehat{x}_0}{dt} = \frac{dx}{dt} - \dot{\sigma}(t)\epsilon$. This decomposition allows $f(\theta_t)$ to represent the noiseless $\widehat{x}_0(t)$ instead of $x(t)$, substantially improving the conditioning of $J$. Fig. 2 illustrates how this separation works in practice.

**Efficient optimization**  The Jacobian $J$ in Eq. 2 represents the derivative of the image with respect to the diffrep parameters. For all but the smallest examples, explicitly forming $J^\top J$ is *computationally intractable*. While iterative linear solvers together with modern automatic differentiation frameworks (e.g., [Potapczynski et al., 2024]) could solve Eq. 2, we found it faster to compute the parameter update as the solution to a non-linear optimization problem. This approach avoids the costly JVP (Jacobian vector product) required to multiply with $J^\top J$.

For each step with size $\Delta t$, we solve:

$$\Delta\theta = \arg\min_{\delta\theta} \| f(\theta) - f(\theta - \delta\theta) - \dot{\sigma}(t)\Delta t(\hat{\epsilon}(f(\theta) + \sigma(t)\epsilon, t) - \epsilon) \|^2. \tag{3}$$

As $\Delta t \to 0$, using the linearization $f(\theta - \Delta\theta) = f(\theta) - J\Delta\theta + \mathcal{O}(\|\Delta\theta\|^2)$, this optimization objective approaches the linear least-squares solution. The optimal $\frac{\Delta\theta}{\Delta t}$ converges to the exact $\frac{d\theta}{dt}$ from Eq. 2. In practice, we employ the Adam optimizer to solve Eq. 3 for whatever discrete $\Delta t$ is used in the diffusion sampling process. This procedure avoids the explicit formation of the linear least-squares solution.

## 3.2   Coupled images, stochastic functions, and 3D rendering

Thus far, we have only considered the case where the output of the render function is a *single image* $x \in \mathcal{X}$. However, our interest lies in render functions $f$ that map parameters $\theta \in \Theta$ to an entire view-dependent image space $\mathcal{H} \subseteq \mathcal{X}^\Pi$. Using the inner product defined in subsection 2.1, we derive the view-dependent pullback PF-ODE:

$$\frac{d\theta}{dt} = f^* \frac{d\widehat{x}_0}{dt} = f^* \left( \frac{dx}{dt} - \dot{\sigma}(t)\epsilon \right) = -\dot{\sigma}(t)\sigma(t)\mathbb{E}\left[ J_\pi^\top J_\pi \right]^{-1} \mathbb{E}\left[ J_\pi^\top \left( \nabla \log p_t(x(\pi), \pi) - \frac{\epsilon(\pi)}{\sigma(t)} \right) \right],$$
$$\tag{4}$$

where the expectations are taken over the views $\pi \sim \Pi$. Here, $\nabla \log p_t(x(\pi), \pi)$ represents the view-specific score function. We use the original score function $\nabla \log p_t(x(\pi))$, with additional view information in the prompt. $\epsilon(\pi)$ denotes the separately managed noise rendered from view $\pi$.

Examining the optimization form of this equation provides further insight. We can interpret it as minimizing the norm in the function space $\left( \| \cdot \|_{\mathcal{H}}^2 = \mathbb{E}_{\pi \sim \Pi} \| \cdot \|^2 \right)$: This is directly analogous to Eq. 3, but with a different inner product. Explicitly written:

$$\Delta\theta = \arg\min_{\Delta\theta} \mathbb{E}_{\pi \sim \Pi} \left[ \| f(\theta, \pi) - f(\theta - \Delta\theta, \pi) - \dot{\sigma}\Delta t(\hat{\epsilon}_t(\pi) - \epsilon(\pi)) \|^2 \right], \tag{5}$$

which we can empirically minimize using view samples $\pi \sim \Pi$.

## 3.3 Consistency and implicit constraints

Our method successfully "pulls back" the PF-ODE $\frac{dx}{dt}$ using the pull-back of the score function $\nabla \log p_t(x(t))$. However, our true goal is to pull back $\nabla \log p_t(x(t)|\hat{x}_0(s) \in \text{range}(f))$ for all $s \leq t$. This ensures that $f$ can render the noiseless components $\hat{x}_0$ throughout the remaining reverse process. For example, when pulling back the PF-ODE for different views of a 3D scene, we want to ensure consistency across all the views.

When $f$ is sufficiently expressive and invertible, $\nabla \log p_t(x(t)|\hat{x}_0(s) \in \text{range}(f)) \approx \nabla \log p_t(x(t))$. This would allow us first to sample $x(0)$ using the PF-ODE from Eq. 1 and then invert $f$ to find $\theta(0)$. However, for most significant applications of our method, $f$ is not invertible.

In noninvertible cases, consider the sampling trajectory $x(t)$ in $\mathcal{X}$ when $\hat{x}_0(t) \notin \text{range } f$, particularly when no nearby sample has high probability in range $f$. Each step of Eq. 2 follows the direction in $\Theta$ that best approximates the direction of the score model in a least-squares sense. When $f$ is not invertible, $J$ is not full rank, and the update to $\theta$ will not precisely match the trajectory in $x$. Consequently, $\| f(\theta_t - \Delta\theta) - x(t - \Delta t) \|$ will be significant. The score model, unaware of this, will continue to point towards high-probability regions outside the range of $f$.

For example, consider an $f$ that only allows low-frequency Fourier modes in generated samples. The unconstrained score model might favor images with high-frequency content (e.g., hair, explosions, detailed textures), resulting in blurry, unresolved images. However, suppose the score model was aware of this constraint against high-frequency details. In that case, it might guide samples toward more suitable content, such as landscapes, slow waves, or impressionist art—dominated by expressible low-frequency components. This issue becomes more pronounced when sampling multiple coupled images through $f$, such as with panoramas and NeRFs, where various views must be consistent.

To address this challenge, we adapt the RePaint method [Lugmayr et al., 2022], initially designed for inpainting, to guide our PF-ODE towards more "renderable" samples. RePaint utilizes the complete Langevin SDE diffusion process, interspersing forward and reverse steps in the sampling schedule. This approach allows the process to correct for inconsistencies during sampling. The forward steps harmonize samples at the current time step by carrying information from earlier steps that implicitly encode the constraint. RePaint requires a stochastic process for conditioning, so we employ the DDIM [Song et al., 2022] sampling procedure with controllable noise at each step. We derive the forward updates of DDIM in Appendix D.

## 3.4 Summary

Our method introduces significant advancements for sampling diffreps using pretrained diffusion models by performing actual sampling instead of merely mode-seeking, thus capturing the full diversity of high-dimensional distributions. We derive the correct pullback of the PF-ODE, incorporating the essential $(J^\top J)^{-1}$ term to ensure unbiased sampling. Efficiency is enhanced through separated noise handling and a faster suboptimization process, allowing practical application for complex representations. Additionally, our approach extends to coupled images and stochastic functions with implicit constraints, making it suitable for tasks like panorama generation and 3D rendering. A comprehensive summary of the contributions in this section and the pseudocode for the DDRep algorithm using DDIM sampling with RePaint is presented in Appendix E.

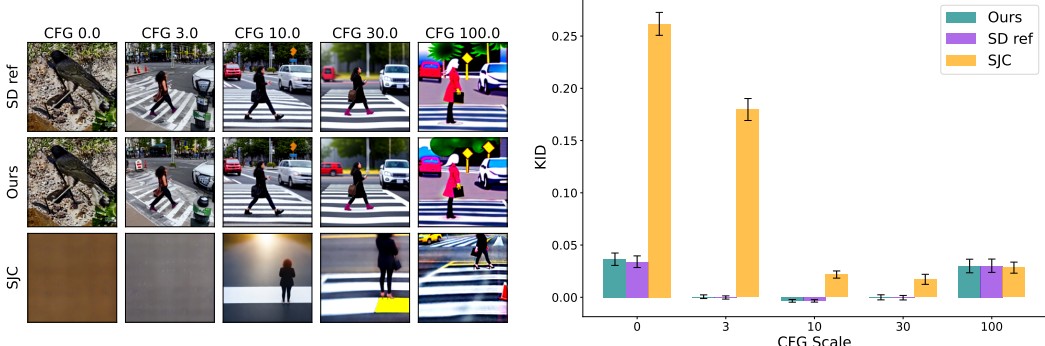

Figure 3: The left figure contains sample renders using the prompt "A woman is standing at a crosswalk at a traffic intersection." from the reference SD (top), our method (middle), and SJC (bottom) over the CFG scales [0,3,10,30,100] from left to right. The right plot is the KID metric (closer to 0 is better) measured on the SIRENs sampled from our method, the SD reference samples, and the SIRENs sampled using SJC.

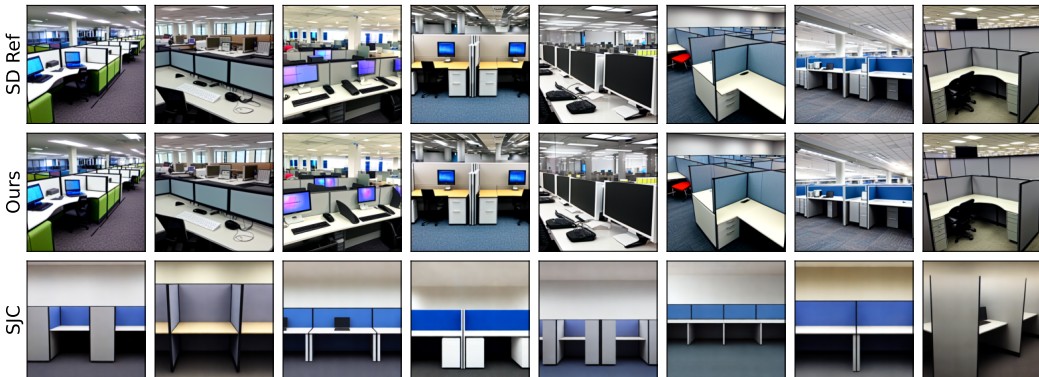

Figure 4: Samples generated using SD ref (top), Our method (middle), and SJC (bottom) using the same prompt "An office cubicle with four different types of computers" with eight different seeds.

## 4 Experiments

All our experiments used the Hugging Face implementation of Stable Diffusion 1.5 (SDv1.5) from Rombach et al. [2022] with the default hyperparameters as the noise predictor model. We conducted our experiments on a single NVIDIA A6000 GPU. We used the complete Langevin SDE given by the DDIM [Song et al., 2022] procedure for all our experiments, with $\eta = 0.75$ as the stochastic interpolation hyperparameter. We interspersed forward and reverse steps to harmonize the diffrep constraints in the sampling procedure. For the suboptimization problem described in Eq. 3, we used the Adam optimizer for 200 steps.

### 4.1 Image SIRENs

We compare the results of our method with those of SJC [Wang et al., 2022] when used to produce SIRENs [Vincent, 2011] for images. While we could directly fit the SIREN to the final image $x(0)$—obtained by first solving the PF-ODE in the image space—we use this scenario to compare different *training-free* generation methods quantitatively.

The SIREN we used was a 3-layer MLP mapping the 2D $(x, y) \in [0, 1]^2$ pixel coordinates to the $\mathbb{R}^4$ latent space of SDv1.5. The MLP had a 128-dimensional sinusoidal embedding layer, a depth of 3, a width of 256, and used $\sin(\cdot)$ as the activation function. To render an image, we used a $64 \times 64$ grid of equally spaced pixel coordinates from $[0, 0]$ to $[1, 1]$ with a shape of $(64, 64, 2)$. The MLP

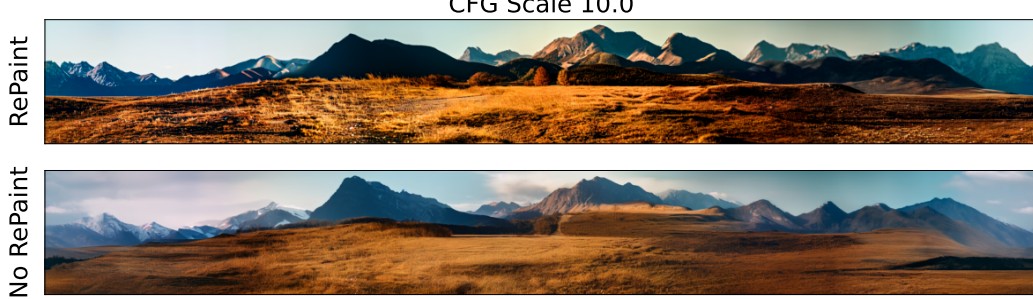

Figure 5: Comparison of landscape panoramas sampled using our method. The top panorama is sampled using the RePaint method, while the bottom is sampled without RePaint. Both approaches use 460 function evaluations (NFEs) to ensure fairness and a CFG scale of 10.0. The prompt for these panoramas was "Landscape picture of a mountain range in the background with an empty plain in the foreground 50mm f/1.8".

output was a latent image with shape $(64, 64, 4)$, which we decoded using the SDv1.5 VQ-VAE into an RGB image with shape $(512, 512, 3)$. We calculated the final metrics on these rendered images.

We used the first 100 captions (ordered by id) from the MS-COCO 2017 validation dataset [Lin et al.] as prompts. We generated three sets of images: (1) reference images using SDv1.5, (2) images rendered from SIRENs sampled using our method, and (3) images rendered from SIRENs generated using SJC [Wang et al., 2022]. We compared these sets against 100 baseline images generated using SDv1.5 with the same prompts but a different seed. We repeated this experiment at five CFG scales [Ho and Salimans] (0, 3, 10, 30, 100). We used the KID metric from Bińkowski et al. to compare the generated images, as it converges within 100 image samples while maintaining good comparison ability. For runtime metrics, see Appendix F.

The example images in Fig. 3 (left) demonstrate that the samples generated by our method are almost indistinguishable from the reference. The results in Fig. 3 (right) clearly show that our method produces images on par with SDv1.5 reference samples.

To further support this observation, we measured the PSNR, SSIM, and LPIPS scores of our method's samples against the SDv1.5 reference images (Table 1). The results show that our method produces SIREN images nearly identical to the reference set.

To demonstrate that our method achieves sampling instead of mode-seeking, we plot eight samples using the same prompt for all models in Fig. 4 with different seeds at CFG scale 10. The samples generated by SJC all look very similar, while those generated by our method and SD show significantly more diversity.

## 4.2 SIREN Panoramas

In this section, we demonstrate how our method can generate SIREN panoramas using the same architecture as the SIREN from subsection 4.1 but with a modified render function. We begin by sampling a view grid of pixel locations with the shape $(8, 64, 64, 2)$ where the batch size is $8$, the height and width are $64$, and the last 2 dimensions are the $(x, y)$ coordinates. The $y$ values are equally spaced in the range [0,1], and $x$ values are equally spaced in the range $[r, r + 1/\texttt{ar}] \mod 1.0$ for $r \sim U(0, 1)$. The $\cdot \mod 1.0$ constraint ensures horizontal wrapping, making the panorama continuous over the entire $360°$ azimuth.

We render the view by sampling from the SIREN at the discrete pixel locations given by the view grid, then decode the latent output using the SDv1.5 VQ-VAE. For our experiments, we assumed that the images were taken by a camera lens with a $45°$ field of view (equivalent to a 50mm focal length), corresponding to an aspect ratio of $\texttt{ar} = 8$.

The SIREN panorama imposes implicit constraints to ensure consistency across overlapping views. Fig. 5 and Fig. 7 illustrate the impact of the consistency conditioning described in subsection 3.3.

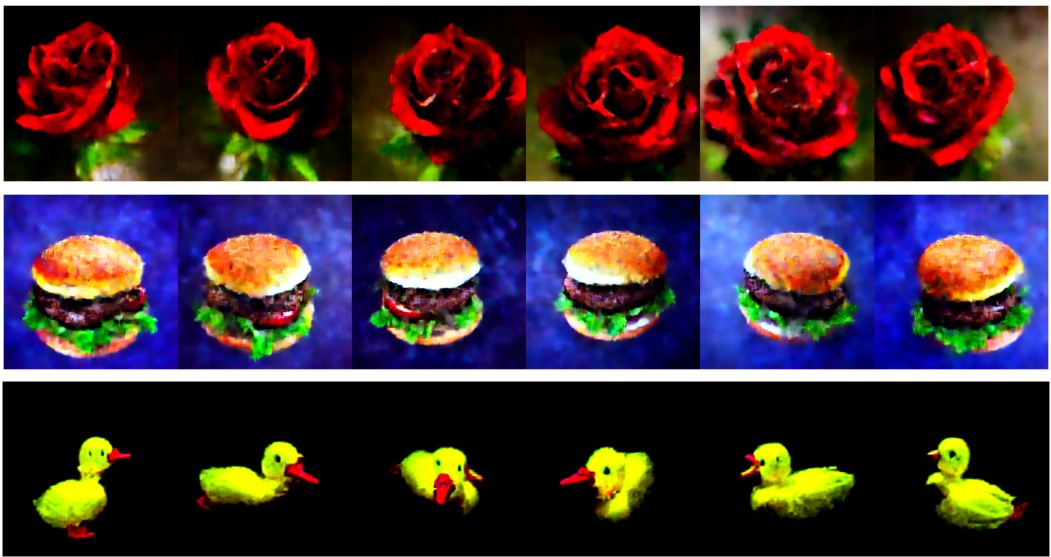

Figure 6: NeRFs generated using our method with the prompts (top) "a photo of a delicious hamburger, centered, isolated, 4K.", (middle) "a DSLR photo of a rose", and (bottom) "a DSLR photo of a yellow duck".

Both RePaint and non-RePaint methods use the same number of function evaluations (460 NFEs) to ensure fair comparisons.

Using RePaint substantially improves the quality and consistency of the generated panorama. This improvement is even more pronounced at low CFG scales, where the diffusion model is encouraged to be more creative and is typically less likely to produce consistent images, as seen in Fig. 7. For runtime metrics and additional panoramas generated using our approach, see Appendix F.

### 4.3 3D NeRFs

For our NeRF experiments, we employed the voxNeRF architecture used by SJC [Wang et al., 2022], chosen for its speed and efficiency. While we did not utilize any additional loss terms from their method, we found that incorporating the object-centric scene initialization from Wang et al. [2023] significantly improved our results. This initialization bootstrapped the diffusion process and led to faster convergence. Specifically, we used the initialization function $\sigma_{\text{init}}(x) = 10\left(1 - \frac{\|x\|}{0.5}\right)$, as detailed in Wang et al. [2023]. For more experimental details, see Appendix F.

The NeRFs generated using our method, as illustrated in Fig. 6, demonstrate our approach's capability to produce detailed, high-quality 3D representations from text prompts. These results showcase the effectiveness of our method in generating complex 3D structures while maintaining consistency across multiple views.

## 5 Conclusion

We have presented a comprehensive, training-free approach for pulling back the sampling process of a diffusion model through a generic differentiable function. By formalizing the problem, we have addressed two critical issues with prior approaches: enabling true sampling rather than just mode seeking (crucial at low guidance levels), and addressing a latent consistency constraint using RePaint to improve generation quality. For future work and limitations, see Appendix G.

Our method opens up new possibilities for generating complex and constrained outputs using pretrained diffusion models. We hope that future research will build on these insights to further expand the capabilities and applications of diffusion models in areas such as 3D content generation, game design, and beyond.

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

# A    Gradient Ascent (GA) does not sample

It is possible to reparameterize the Gradient Ascent (GA) procedure from Poole et al. [2022] and Wang et al. [2022] in a way that resembles solving the PF-ODE by carefully selecting optimizer hyperparameters and stopping times. However, this reparameterization is **highly specific**. The crucial difference lies in the interpretation of the trajectory: in GA, the path taken is typically *irrelevant* as long as it converges to the critical point, allowing us the flexibility to multiply the gradient by any positive semidefinite (PSD) matrix without affecting the solution.

For the PF-ODE, however, the **trajectory is everything**. The stopping time, weighting terms, and score evaluations must be precisely aligned to ensure that the solution looks like a typical sample. Even small deviations in the path, like evolving the PF-ODE for too long or too short, can lead to either *oversmoothed* samples (with too much entropy removed) or *under-resolved*, noisy samples. Thus, while GA might superficially look like a discretized version of the PF-ODE, the underlying processes are inherently distinct: one finds the mode, and the other describes the full trajectory required to generate samples from the distribution. In the case of SDS and SJC, they both use Adam as the optimizer, ignoring any of the nuanced considerations needed to transform the PF-ODE into the corresponding optimization procedure.

In practice, this distinction means that while both SDS and SJC may yield plausible diffreps at *high classifier-free guidance levels*, they tend to falter when producing coherent samples under lower CFG weights–particularly for complex, multimodal distributions with many degrees of freedom. High CFG levels may mask these limitations by strongly biasing the output towards the mode, which aligns more closely with the GA approach, but at the cost of **reduced sample diversity** and potentially **missed details** captured by the full distribution.

**The problem with mode-seeking**    To appreciate the limitations of using the mode as a proxy for sampling, we consider the multi-modal and the high-dimensional settings.

In the **multimodal** setting, the sample distribution may contain several distinct high-density regions. However, mode-seeking algorithms focus on only one of these regions, sacrificing sample diversity.

In low-dimensional spaces, the mode often resembles a typical sample. However, this intuition breaks down in **high-dimensional** spaces ($d \gg 1$), such as the space of images. This counterintuitive behavior can be explained using the *thin-shell phenomenon*. For a high-dimensional standard Gaussian distribution, samples are not concentrated near the mode (the origin), as one might expect. Instead, they predominantly reside in an exponentially thin shell at the boundary of a $\sqrt{d}$-radius ball centered on the mode. This phenomenon explains why the mode can be an anomalous and atypical point in high-dimensional distributions.

As an illustrative example, consider sampling a normalized "pure-noise" image. Despite having a mode of $0$, we would almost never expect to sample a uniformly gray image. This provides some insight as to why the mode of a distribution with several degrees of freedom will lack the quality and details present only in samples from the thin shell.

# B    General conditioning

While RePaint can be used to condition the reverse trajectory, it tends to be less generalizable for several tasks. For more general conditioning—including inverse problems—MCG [Chung et al., 2022] and its extension DPS [Chung et al., 2023] have gained popularity. These methods add a corrective term to the score, encouraging reverse steps to minimize constraint violations for the predicted final sample $\hat{x}_0$ according to the Tweedie formula. MCG and DPS have seen considerable application and extension to broader settings [Bansal et al., 2023]. Finzi et al. [2023] showed this correction to be an asymptotically exact approximation of $\nabla \log p_t(\text{constraint}|x)$, while Rout et al. [2023] derive the relevant term from Bayes' rule and higher-order corrections.

# C    PF-ODE Pullback Derivation and Geometric Details

This section provides the details for the derivation of Eq. 2 using differential geometry.

First, we provide a full expansion of Eq. 1. For any point in our sample space $x \in \mathcal{X}$, we can define a Riemannian metric $M_x$ that describes the local geometry of the space around $x$. Given tangent vectors $y, z$ at point $x$, we define the inner product using this metric $\langle y, z \rangle_{M_x} = y^T M_x z$ where $M_x$ is represented using a positive-definite matrix. We implicitly use the Euclidean metric for the image space $\mathcal{X}$, which corresponds to the identity matrix $M_x = I_d, \forall x \in \mathcal{X}$. However, in curved spaces such as the parameter space $\Theta$, the corresponding Riemannian metric is not necessarily the identity and can vary based on where it is evaluated.

Here, we use matrix notation rather than Einstein summation or coordinate-free notation for familiarity within the machine learning context. However, we note that matrix notation can obscure some important aspects of differential geometry. For example, while the components of the Euclidean metric in the Euclidean coordinate basis form the identity matrix, the metric is not the identity function, as it transforms between two distinct vector spaces. As such, when using the matrix notation, it becomes very important to keep track of the types of objects rather than just their values.

Because we use the Euclidean metric for the image space, its role in Eq. 1 is hidden. Explicitly incorporating the metric into the process, we get the following expanded metric PF-ODE

$$\frac{dx}{dt} = -\dot{\sigma}(t)\sigma(t)M_x^{-1}\nabla \log dP_t/d\lambda_x, \tag{6}$$

where $P_t$ is the probability measure at time $t$ and $\lambda_x$ is the Lebesgue measure on $\mathcal{X}$, and $dP_t/d\lambda_x$ is the Radon-Nikodym derivative. Notice that when we use $M_x = I_d, \forall x \in \mathcal{X}$, this equation is equivalent to Eq. 1. If we pull back the dynamics of this process through the render function $f : \mathcal{X} \to \Theta$ to the parameter space $\Theta$, we must pull back *both* the score function and the metric $M_x$ to get the pulled back the vector field $f^* \frac{dx}{dt}$. The different object types transform as follows with the pullback $f^*$:

| | | |
|---|---|---|
| Scalar Functions | $s \in C(\mathcal{X})$ : | $(f^*s)(\theta) = s(f(\theta))$ | (7) |
| Cotangent Vectors | $\omega \in T^*\mathcal{X}$ : | $(f^*\omega)(\theta) = J^\top \omega(f(\theta))$ | (8) |
| Metric Tensor | $M \in T^*\mathcal{X} \otimes T^*\mathcal{X}$ : | $(f^*M)(\theta) = J^\top M_{f(\theta)} J$ | (9) |
| Tangent vectors | $v \in T\mathcal{X}$ : | $(f^*v)(\theta) = (J^\top M_{f(\theta)} J)^{-1} J^\top M_{f(\theta)} v(f(\theta))$, | (10) |

where $J$ is the Jacobian of $f$ evaluated at $\theta$. For a scalar function $s(x)$, the gradient $\nabla s(x)$ is *co*-vector field (also known as a differential 1-form) and can be converted into a vector field using the inverse metric: $M_x^{-1}\nabla s(x)$. We can see that Eq. 10 can be expressed concisely in terms of the pullback metric $f^*v = (f^*M)^{-1}f^*(Mv)$, which geometrically corresponds to converting the vector field to a covector field with $M$, pulling back the covector field (using the chain rule), and then converting back to a vector field with the inverse of the pulled back metric. This sequence is illustrated in Fig. 1.

From Eq. 9, the pullback of the Euclidean metric is $J^\top I J = J^\top J$. From these expressions, we can see that the form of $f^* \frac{dx}{dt}$ should be:

$$f^*\frac{dx}{dt} = -\dot{\sigma}(t)\sigma(t)(J^\top M_{f(\theta)} J)^{-1} J^\top M_{f(\theta)} M_{f(\theta)}^{-1} \nabla\big(\log dP_t/d\lambda_x\big)$$

$$f^*\frac{dx}{dt} = -\dot{\sigma}(t)\sigma(t)(J^\top M_{f(\theta)} J)^{-1} \nabla\big(\log dP_t/d\lambda_x\big)$$

$$f^*\frac{dx}{dt} = -\dot{\sigma}(t)\sigma(t)(J^\top J)^{-1} \nabla \log p_t \qquad \text{(on Euclidean } \mathcal{X} \text{ where } M_x = I).$$

Thus the pullback of the reverse time PF-ODE is

$$\frac{d\theta}{dt} = f^*\frac{dx}{dt} = (J^\top J)^{-1} J^\top \frac{dx}{dt} = -\dot{\sigma}(t)\sigma(t)(J^\top J)^{-1} J^\top \nabla \log p_t(f(\theta)). \tag{11}$$

**Change of variables contribution.** Some readers may be surprised to see that the Jacobian log determinant *does not* show up in this transformation. Though somewhat technical, this can be seen by unpacking the Radon-Nikodym derivative $dP_t/d\lambda_x$. In general, on a manifold with metric $M$, the Lebesgue measure is given by $d\lambda = \sqrt{\det M_x}dx$. Therefore, when written in terms of the density $dP_t/dx = p_t(x)$, one has $dP_t/d\lambda_x = p_t(x)/\sqrt{\det M_x}$. When evaluated only in the Euclidean space

where the coordinates are chosen such that $M_x = I$, this detail can be safely ignored. However, when pulling back to the parameter space, these hidden terms matter. The Radon-Nikodym derivative $dP_t/d\lambda_x$ is a scalar field and is not transformed under the pullback, however $p_t$ is a scalar density and satisfies

$$(f^*p_t)(\theta) = p_t(f(\theta))\sqrt{\det(J^\top J)},$$

the familiar change of variables formula. When the input and output dimensions are the same $p_\Theta(\theta) = p_\mathcal{X}(f(\theta))\det J$. However, $\sqrt{\det M_x}$ is also a scalar density and similarly transforms:

$$f^*\sqrt{\det M_x} = \sqrt{\det f^*M_x} = \sqrt{\det J^\top M_x J} = \sqrt{\det J^\top J}.$$

Assembling these two components together, we see that the determinant contribution from the change of variables formula cancels with the contribution from the change of the Lebesgue measure in the sampling ODE.

$$
\begin{aligned}
f^*\nabla \log(dP_t/d\lambda_x) &= f^*\nabla \log\left(p_t/\sqrt{\det M_x}\right) \\
f^*\nabla \log(dP_t/d\lambda_x) &= J^\top\nabla\left(\log(f^*p_t) - \log f^*\sqrt{\det M_x}\right) \\
f^*\nabla \log(dP_t/d\lambda_x) &= J^\top\nabla\left(\log p_t + \frac{1}{2}\log\det J^\top J - \frac{1}{2}\log\det J^\top M_x J\right) \\
f^*\nabla \log(dP_t/d\lambda_x) &= J^\top\nabla \log p_t.
\end{aligned}
$$

This is why the change of variables term $\frac{1}{2}\log\det J^\top J$ term does not appear in the pullback of the PF-ODE.

## D   Details on constrained sampling

For our experiments, we used the stochastic non-Markovian reverse process from DDIM Song et al. [2022]. For the RePaint steps, we also need to take forward steps to encourage the samples that are "renderable". In this section, we prove the correct form for the forward process looks like

$$x(t) = s(t)\widetilde{x}_0(t - dt) + \frac{\sigma(t)}{\sigma(t - dt)}\left(\sqrt{\sigma^2(t - dt) - \tau^2(t)}\widetilde{\epsilon}_0(t - dt) + \tau(t)\epsilon_f(t)\right). \qquad (12)$$

Where $\sigma(t)$ is the standard deviation and can be calculated using $\sigma(t) = \frac{1}{\sqrt{1/\alpha^2(t)+1}}$ and $s(t)$ is the scale of signal. It can be calculated using $s(t) = \frac{1}{\sqrt{\alpha^2(t)+1}}$ where $\alpha(t)$ is the noise-to-signal ratio. In the main body of the text, we let $\sigma(t)$ notate the noise-to-signal ratio, and we assume $s(t) = 1$ but that is not necessarily true in the general case.

**Theorem D.1.** *Eq. 12 is the correct forward update for the non-Markovian process.*

*Proof.* We will use the notation $x(t|0) = x(t) = x(t)|x(0), x(t|t - dt, 0) = x(t)|x(t - dt), x(0)$, and $x(t - dt|t, 0) = x(t - dt)|x(t)x(0)$. The same holds if we replace $x$ with $\epsilon_0$.

Using the reparametrization trick we can write $x(t) = x(t|0) = s(t)x(0) + \sigma(t)\epsilon_0(t|0)$, so

$$\ln p(x(t|0)) = -\frac{1}{2}\epsilon_0^2(t|0) + c.$$

Using DDIM and the reparameterization trick, we can also write

$$\ln p(x(t - dt|t, 0)) = -\frac{1}{2}\left(\frac{1}{\tau(t)}x(t - dt) - \frac{s(t - dt)}{\tau(t)}x(0) - \frac{\sqrt{\sigma^2(t - dt) - \tau^2(t)}}{\tau(t)}\epsilon_0(t)\right)^2 + c$$

$$= -\frac{1}{2}\left(\frac{\sigma(t - dt)}{\tau(t)}\epsilon_0(t - dt) - \frac{\sqrt{\sigma^2(t - dt) - \tau^2(t)}}{\tau(t)}\epsilon_0(t)\right)^2 + c$$

Now we can use Bayes' theorem

$$p(x(t|t-dt,0)) = \frac{p(x(t-dt|t,0))p(x(t|0))}{p(x(t-dt|0))}$$

$$\ln p(x(t|t-dt,0)) = -\frac{1}{2}\left(\left(\frac{\sigma(t-dt)}{\tau(t)}\epsilon_0(t-dt) - \frac{\sqrt{\sigma^2(t-dt)-\tau^2(t)}}{\tau(t)}\epsilon_0(t)\right)^2 + \epsilon_0^2(t) - \epsilon_0^2(t-dt)\right) + c$$

$$= -\frac{1}{2}\left(\frac{\sigma(t-dt)}{\tau(t)}\epsilon_0(t) - \frac{\sqrt{\sigma^2(t-dt)-\tau^2(t)}}{\tau(t)}\epsilon_0(t-dt)\right)^2 + c$$

$$= -\frac{1}{2}\cdot\frac{\sigma^2(t-dt)}{\tau^2(t)}\left(\epsilon_0(t) - \frac{\sqrt{\sigma^2(t-dt)-\tau^2(t)}}{\sigma(t-dt)}\epsilon_0(t-dt)\right)^2 + c$$

Using reparameterization, we can say that (conditioned on $x(0)$), $x(t|t-dt,0) = s(t)x(0) + \sigma(t)\epsilon_0(t|t-dt,0)$, where

$$\epsilon_0(t|t-dt,0) \sim \mathcal{N}\left(\frac{\sqrt{\sigma^2(t-dt)-\tau^2(t)}}{\sigma(t-dt)}\epsilon_0(t-dt), \frac{\tau^2(t)}{\sigma^2(t-dt)}\right)$$

$\square$

## E  Summary of key ideas and algorithm

Our method introduces several crucial innovations for sampling differentiable representations using pretrained diffusion models:

1. **True sampling vs. mode-seeking:** Unlike SJC and SDS, which primarily find modes of the distribution, our method aims to generate true samples. This distinction is critical for capturing the full diversity and characteristics of the underlying distribution, especially in high-dimensional spaces.

2. **Correct pullback of the PF-ODE:** We derive the mathematically correct form of the pulled-back Probability Flow ODE, which includes a crucial $(J^\top J)^{-1}$ term. While this term acts as a preconditioner in optimization-based approaches (affecting only convergence rates), it is essential for unbiased sampling in our PF-ODE framework. Omitting this term leads to incorrect results.

3. **Efficient implementation:** We introduce techniques for efficient implementation, including separated noise handling and a faster suboptimization procedure. These innovations allow for the practical application of our method to sample complex differentiable representations.

4. **Generalization to coupled images and stochastic functions:** Our method extends naturally to scenarios involving coupled images and stochastic functions, making it applicable to a wide range of problems, including 3D rendering and panorama generation.

5. **Handling multimodality and constraints:** Our sampling approach naturally handles multimodal distributions, unlike mode-seeking methods. However, this introduces challenges when the samples are incompatible with the constraints imposed by the differentiable representation. We address this by conditioning the sampling process with the consistency constraint using an adapted RePaint method.

These key ideas collectively enable our method to generate high-quality, diverse samples of differentiable representations by encouraging the diffusion model to maintain consistency with the constraints imposed by the representation.

The pseudocode for the DDRep algorithm using DDIM sampling with RePaint is presented in Alg. 1.

## F  Additional experimental details

This section presents additional details and runtime metrics for our algorithm with the experimental settings given in section 4.

**Algorithm 1** DDRep with RePaint

1: **Input:** cfg, reverse_steps, jump_interval, jump_len, jump_repeat, $\eta$
2: **Init:** $\theta = \arg\min_\theta \mathbb{E}_{\pi \sim \Pi} \| f(\theta, \pi) \|$, $\epsilon(\pi) \sim \mathcal{N}(0, I), \forall \pi \in \Pi$
3: schedule = RePaintSchedule(reverse_steps, jump_interval, jump_len, jump_repeat)
4: **for** (reverse, $t$) **in** schedule **do**
5:     $\sigma_{\text{langevin}} = \eta \sqrt{\sigma^{-2}(t) + 1} \sqrt{1 - \frac{\sigma^2(t - \Delta t) + 1}{\sigma^2(t) + 1}}$
6:     $\epsilon_{\text{langevin}} \sim \mathcal{N}(0, \sigma_{\text{langevin}}^2)$
7:     **Sample Views:** $\pi \sim \Pi$
8:     **if** reverse **then**
9:         $x(t) = f(\theta, \pi) + \sigma(t)\epsilon(\pi)$
10:         $\hat{\epsilon}_t(\pi) = \hat{\epsilon}(x(t), t, \text{view\_prompt}(\pi), \text{cfg})$
11:         $\widehat{x}_{\text{next}} = x(t) - \sigma(t)\hat{\epsilon}_t(\pi) + \sqrt{1 - \sigma_{\text{langevin}}^2}\sigma(t - \Delta t)\hat{\epsilon}_t(\pi) + \sigma(t - \Delta t)\epsilon_{\text{langevin}}$
12:     **else**
13:         $\widehat{x}_{\text{next}} = f(\theta, \pi) + \sigma(t)\sqrt{1 - \sigma_{\text{langevin}}^2}\epsilon(\pi) + \sigma(t)\epsilon_{\text{langevin}}$
14:     **end if**
15:     **Update Noise Parameters:** $\epsilon(\pi) = \sqrt{1 - \sigma_{\text{langevin}}^2}\epsilon(\pi) + \epsilon_{\text{langevin}}$
16:     **Update Diffrep Parameters:** $\theta = \arg\min_\theta \| f(\theta, \pi) - \widehat{x}_{\text{next}} + \sigma_{\text{next}}\epsilon(\pi) \|$
17: **end for**

| cfg scale | 0 | 3 | 10 | 30 | 100 |
|---|---|---|---|---|---|
| PSNR ↑ | 29.712 ± 4.231 | 29.931 ± 5.767 | 27.593 ± 6.574 | 23.453 ± 6.397 | 13.586 ± 4.024 |
| SSIM ↑ | 0.899 ± 0.086 | 0.896 ± 0.107 | 0.888 ± 0.115 | 0.826 ± 0.145 | 0.523 ± 0.183 |
| LPIPS ↓ | 0.044 ± 0.050 | 0.053 ± 0.074 | 0.061 ± 0.074 | 0.098 ± 0.098 | 0.336 ± 0.138 |

Table 1: Image similarity scores (mean ± std) for the images produced using our method and those produced using the SD reference.

**Image SIRENs**  Sampling a batch of eight reference images using SDv1.5 takes 39 seconds. Generating SIRENs using SJC depends on the number of iterations used; running SJC for 3000 iterations to generate eight SIRENs takes 694 seconds, corresponding to a per-iteration time of 2.31 seconds. In comparison, sampling a batch of eight SIRENs using our method takes 82 seconds. These timings demonstrate that our method offers a significant speed advantage over SJC while producing comparable results to the reference SDv1.5 samples.

The PSNR, SSIM, and LPIPS scores of our method against the SDv1.5 reference images are presented in Table 1.

**SIREN Panoramas**  Generating a SIREN panorama with our method takes 218 seconds. Fig. 7 shows how using RePaint to condition the sampling procedure can generate more coherent panoramas for low CFG scales. Additional examples of SIREN panoramas generated using our method are provided in Fig. 8.

**3D NeRFs**  Our algorithm processed a batch of eight views per iteration, which was the maximum capacity for the A6000 VRAM. The generation of each NeRF required approximately 7.2 seconds per NFE (Number of Function Evaluations). We implemented 100 inference steps using the DDIM procedure, with one RePaint forward step for each reverse step. This configuration resulted in 199 NFEs per NeRF, translating to about 24 minutes of sampling time for each NeRF. It is worth noting that run times can fluctuate based on the specific sampling procedure and GPU architecture used. For comparison, generating a comparable NeRF using SJC takes approximately 34 minutes for 10,000 steps, based on the code provided by the authors.

CFG Scale 3.0

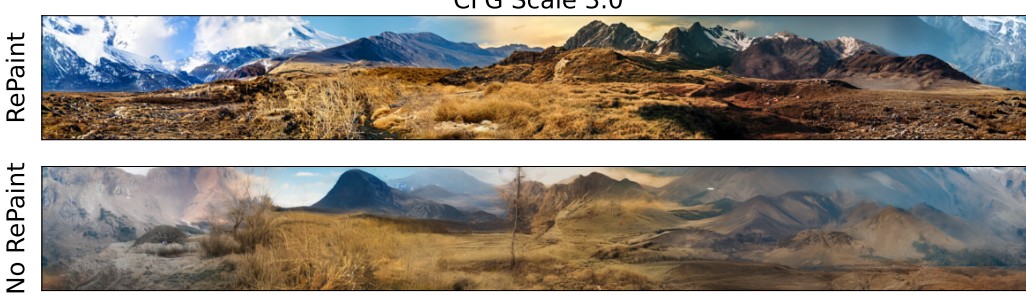

Figure 7: Comparison of landscape panoramas sampled using our method. The top panorama is sampled using the RePaint method, while the bottom is sampled without RePaint. Both approaches use 460 function evaluations (NFEs) to ensure fairness and a CFG scale of 3.0. The prompt for these panoramas was "Landscape picture of a mountain range in the background with an empty plain in the foreground 50mm f/1.8".

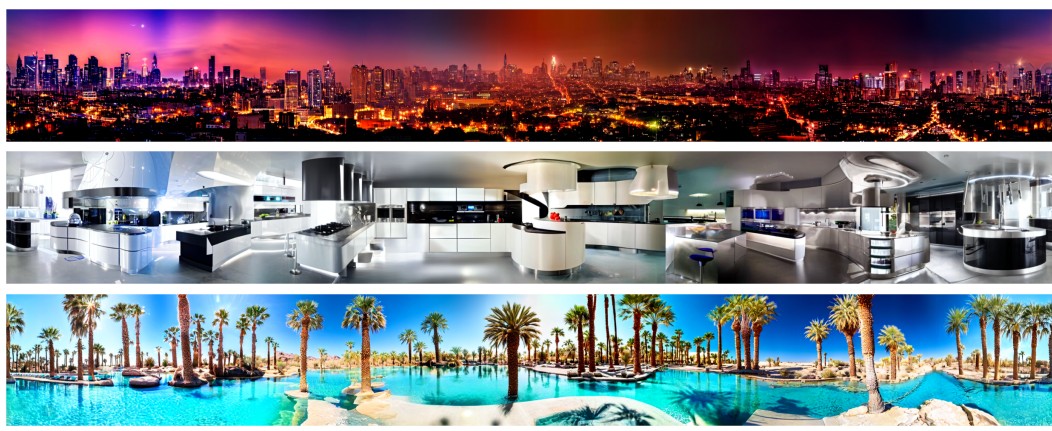

Figure 8: Additional SIREN Panoramas with prompts (top) "Urban skyline at twilight, city lights twinkling in the distance." (middle) "A futuristic kitchen" (bottom) "Desert oasis, palm trees surrounding a crystal-clear pool."

## G   Future work and limitations

Our work has demonstrated the zero-shot generation of implicit images, panoramas, and NeRFs. The versatility of our approach opens up a wide range of potential applications, including vector graphics generation, embedding diverse content in scenes through geometric transformations, differentiable heightmaps, and applications using differentiable physics renderers.

While we focused on NeRFs in this paper, more advanced differentiable representations like Gaussian splats and full-scene representations have emerged. Our method should be applicable to these cases, but further work is needed to adapt our approach to these newer representations.

Looking ahead, one could envision generating entire maps or game environments using diffusion models. However, this would likely require innovations in amortization and strategies for splitting and combining subproblems. In these cases, the function $f$ may be even more restrictive than for SIRENs and NeRFs, underscoring the importance of conditionally sampling the implicit constraint.

**Limitations**   Our method faces two significant limitations. First, the additional steps required for RePaint introduce computational overhead. A priori, we cannot determine how many steps are needed to harmonize the constraint into the diffrep. Transitioning to a conditional sampling method like MCG could potentially allow us to skip these extra forward steps and directly integrate the constraint into the solver. However, this requires further investigation to ensure compatibility with our method.

Second, the substantial stochasticity in the Monte Carlo estimate of the pullback score over several views, particularly when the Jacobian of the render function is ill-conditioned, poses a challenge. While increasing the number of sample views could reduce variance in our estimate, it would also slow down our algorithm and consume more memory. Further research into view sampling techniques, such as importance sampling, could help decrease variance without compromising computational efficiency.

