# OpenReview forum: "Diffusing Differentiable Representations"
_NeurIPS.cc/2024/Conference — NeurIPS 2024 poster_

### Official Review · Reviewer_za3b · 2024-07-07

**Soundness:** 4
**Presentation:** 3
**Contribution:** 4
**Rating:** 6
**Confidence:** 3

**Summary:**

This paper introduces a zero-shot method to sample neural representations with pre-trained diffusion models. By pulling back the measure over the data space through the representation, the authors express the PF ODE in the parameter space. Solving this ODE can directly provide parameter samples of the representation. The authors also discuss cases where the representation needs to model coupled images. In this scenario, the PF ODE is expressed by expectation and can be solved following the same principle. This method offers random samples rather than returning the mode, which can yield results with higher diversity compared to baselines without sacrificing performance. The method is also compatible with RePaint, which can ensure the implicit constraint (e.g., the inductive bias of the representation or practical consistency constraint).

**Strengths:**

1. This work introduces a zero-shot method to sample neural representations by pre-trained diffusion models. The method is fully zero-shot and does not rely on the specific form of the representation. Therefore, it may gain great potential in downstream tasks where training is difficult, or the representation is highly restrictive.
2. This method can generate samples with significantly higher diversity compared to the baselines, and can ensure consistency constraint using RePaint, improving global coherence and quality of generation.
3. The idea to handle the Jacobian matrix by approximating each ODE step using an optimization task is cute and neat.

**Weaknesses:**

My primary concern is its runtime. If I understand correctly, you need to run the optimization for several iterations for each data point and each Euler step. And for tasks like NeRF, where the PF ODE involves expectations, this optimization also involves an MC estimator. However, this paper only reported the runtime for 3D NeRF experiments. I am curious on

a. what is the runtime for other experiments? What is the runtime for the baselines? Is this method significantly more expensive than others?

b. How many samples do you use to estimate the expectation? Does it contribute to the runtime? Will reducing the number of samples increase the variance and lead to suboptimal performance?

I would still appreciate this approach even though it may be more expensive. However, reporting the runtime and comparing it with the previous baselines can help us understand this method better.

**Questions:**

I have two more questions regarding the experiments.
1. In Fig 1 (right), you compare your method with SJC and claim that "the samples produced by our method are almost indistinguishable from the reference." However, I do not necessarily agree with this claim. Your method yields much smoother images than SD. Even though I mainly suspect this may be due to SIREN's inductive bias, it is visible and distinguishable. Additionally, it seems that your method provides very different outcomes with CFG 30, while results by CFG 3/10/100 are pretty similar. Do you have any explanation for this? Is it a sign that this approach is not robust?
2. The PF ODEs in all experiments need to meet some implicit constraints due to the inherent bias of INRs. However, you only apply RePaint for Panorama SIRENs to ensure the consistency of the generated panorama. Do I understand this correctly? Is there a difference if you ensure the implicit constraints for other experiments? Is it possible to compare the results with and without ensuring the constraint?


In general, I appreciate the contribution of this work and would be happy to raise my score if my questions are adequately addressed.

**Limitations:**

I did not find discussions on limitations even though Sec 6 is called Limitations and Conclusion. I am unsure if this work has no limitations, such as runtime. Could the author please clarify this?

---

> ### Author Rebuttal · Authors · 2024-08-07
>
> Thank you for your comments and review. We will address your concerns in turn.
>
> Experiment details: As mentioned in the paper, generating a NeRF with our method takes around 24 minutes for 199 NFEs. We used as many samples as possible in the 40GB VRAM of an NVIDIA A6000; our experiments were eight samples. Generating a comparable NeRF using SJC takes around 34 minutes for 10000 steps using the code they provide. Generating eight batched SIREN images using our method takes 82 seconds. Generating a SIREN Panorama with our method takes 218 seconds.
>
> As you correctly pointed out, Figure 1 in the paper had some issues (the images were out of order). We have briefly described the problems in the common response and attached the corrected Figure 1 in the accompanying PDF.
>
> We use RePaint for all our experiments, including the SIREN image and NeRF experiments. Without the RePaint method, the results are much less compelling. In fact, the NeRF experiments diverge without it. The INR implicitly defines the constraint, so it is impossible to remove the constraints from the INR. However, as a proxy for the constraint, one could consider using the CFG scale, where a high CFG corresponds to a stronger constraint because it enforces the renders from the INR to look a specific way. On the other hand, low CFG means more flexibility in the kinds of INR we can sample. Figure 3 shows the results of ablating RePaint on SIREN Panoramas with a high and a low CFG.
>
> We thank you for your time spent reviewing. We have invested substantial effort in addressing your concerns and improving the quality of the paper, and we ask that you consider adjusting your score in light of our response. Please let us know if you have additional questions we can help answer.

---

> ### Comment · Reviewer_za3b · 2024-08-08
>
> Thank you for your reply and for providing more details. However, I have 3 more further questions:
>
>
> 1. How many iterations per Eular step do you use for optimization?
>
> 2. >  Generating eight batched SIREN images using our method takes 82 seconds. Generating a SIREN Panorama with our method takes 218 seconds.
>
> What is the baseline methods' runtime?
>
> 2. My concern regarding limitations has not been addressed yet. Did you discuss limitations in Section 6? Or do you think there is no clear limitation?

---

> > ### Author Response · Authors · 2024-08-09
> >
> > Thank you for your comment.
> >
> > In our experiments, we used 100 inference (reverse, Euler) steps of DDIM and a single forward step for RePaint at every increment except the first one. This schedule corresponds to 199 steps in total. We optimize the INR using 200 Adam steps for each inference step.
> >
> > Generating a batch of 8 reference images takes 39 seconds. Generating 8 SIREN images using SJC depends on the number of iterations used. Running SJC for 3000 iterations to generate 8 SIRENs takes 694 seconds, corresponding to a per-iteration time of 2.31 seconds. Generating a non-SIREN panorama with eight views using our method takes around 41 seconds, which makes sense since this is essentially the same problem as generating eight batched images.
> >
> > We highlight some specific limitations from Section 6 and include some additional ones here:
> >
> > - One significant limitation of our method is the additional steps we need for RePaint. A priori, we do not know how many steps are required to harmonize the constraint into the diffrep. Moving to a conditional sampling method like MCG would allow us to skip the extra forward steps and directly integrate the constraint into the solver. However, this needs further exploration to work with the rest of our method.
> > - There is also often significant stochasticity in the Monte Carlo estimate of the pullback score over several views, especially when the Jacobian of the render function is ill-conditioned. We would prefer to take more sample views to reduce the variance in our estimate, but this slows down our algorithm or takes more memory. More work is needed to improve the view sampling techniques (like importance sampling) to decrease the variance in our estimates without slowing down our algorithm.
> > - Finally, we only investigated our method on NeRFs; since then, there have been significantly more advanced diffreps like Gaussian Splats and full-scene representations. While our method should also work for these cases, more work needs to be done to delineate the specifics of how to adapt our method for these cases.
> >
> > We will add all these details to the final version of the paper.

---

> > > ### Comment · Reviewer_za3b · 2024-08-09
> > >
> > > I feel my concern has already been addressed. Therefore, I raise my rating for this work.

---

### Official Review · Reviewer_wR2k · 2024-07-11

**Soundness:** 2
**Presentation:** 2
**Contribution:** 2
**Rating:** 4
**Confidence:** 2

**Summary:**

The paper introduces a method for sampling a differential representation using a pre-trained diffusion model. Instead of sampling in the image space, the authors propose sampling in the parameter space of differential representations by 'pulling back' the probability distribution from the image space to the parameter space of the differential representation.

**Strengths:**

- The paper introduces a novel method for sampling a differentiable representation for a pre-trained diffusion model, which is highly relevant.
- Their method appears to be well-principled.
- Although their results are limited, they seem promising.

**Weaknesses:**

- The technical part of the paper is difficult to follow.
- Comparison to the state-of-the-art is provided only in Section 5.1, where a differential representation is not required.
- The code is not shared at this stage.

**Questions:**

- The two main relevant papers (Poole et al. [2022] and Wang et al. [2022]) are not discussed in appropriate detail:
  - The authors claim equivalence between the methods without providing a detailed argument or a citation where the argument is made.
  - Lines 93-94 suggest that these methods act as "mode-finding algorithms," however, a discussion is lacking.

- The arguments in Section 3.1 (which I believe is the main argument of the paper) are hard to follow for a non-expert:
  - What is the relevance of Equations 3-5?
  - How are Equations (6) and (7) derived?

**Limitations:**

yes

---

> ### Author Rebuttal · Authors · 2024-08-07
>
> Thank you for your comments and review. We will address your concerns in turn.
>
> SJC and Dreamfusion (SDS) are extremely similar in methodology when accounting for the conversion between the denoising model and the score function. The key differences are in the weighting function that the expectation is taken over, the optimization and sampled times in the schedule, and regularizers added for the 3D case. We show how the SJC and SDS objectives are identical up to a weighting term and delineate the major differences between the methods and how they compare to our method in the common response.
>
> Reliable metrics for generative models are primarily image-based. This is why we used the SIREN image diffrep to evaluate our method quantitatively. While it is possible to generate NeRFs and SIREN panoramas using the other methods, comparing them would be infeasible since there are no good techniques to compare distributions of general diffreps yet.
>
> We will make the code public in the final version of the paper.
>
> We have significantly changed the technical description of our methodology to make it simpler and more intuitive. We hope you find this new description clearer.
>
> The score model associated with the noise predictor implicitly defines a distribution on the data space $\mathcal X$, and the reverse time PF-ODE provides a means to sample from that distribution.
> In this section, we will show how to use the score model and the PF-ODE to sample the parameters of a differentiable representation (diffrep) so that the rendered views look like samples from the implicit distribution.
>
> In SJC and DreamFusion, they derive their parameter updates using the pullback of the sample score function through the render map $f$. Explicitly, this pullback is obtained by using the chain rule $\tfrac{d(\log p)}{d\theta} = \tfrac{d(\log p)}{dx}\tfrac{dx}{d\theta}$. The pullback score is then used __as is__ to perform gradient ascent in the parameter space. However, since $p$ is a distribution and not a simple scalar field, the change of variables formula requires an additional $\log \det J$ for the appropriate score function in the parameter space, where $J$ is the Jacobian of $f$. Carefully examining this approach through the lens of differential geometry reveals even deeper issues.
>
> From differential geometry, recall that it is a vector field (a section of the tangent bundle $T\mathcal X$), not a differential form (a section of the cotangent bundle $T^*\mathcal X$), which defines the integral curves (flows) on a manifold. To derive the probability flow ODE in the parameter space, we must pull back the vector field $\frac{dx}{dt}\in T\mathcal X$ to the parameter vector field $\frac{d\theta}{dt} \in T\Theta$ using the render map $f$. The pullback of the vector field through  $f$ is given by
> $(f^* \frac{dx}{dt})|_\theta = (J^\top J)^{-1}J^\top \frac{dx}{dt}|\_{f(\theta)},$ and thus the pullback of the probability flow ODE is
>
> \begin{equation}
>     \frac{d\theta}{dt} = -\dot\sigma(t)\sigma(t)(J^\top J)^{-1}J^\top\nabla \log p_t(f(\theta)).
> \end{equation}
>
> We note that this is in contrast to the SJC and DreamFusion score. The confusion arises when comparing the types of elements that are pulled back. While the left-hand side of the PF-ODE $\frac{dx}{dt}$ is a vector field, the score function $\nabla \log p_t(x(t))$ on the right-hand side is a covector field, i.e., it is a differential form. Pulling back the score function as a differential form correctly yields $J^\top\nabla \log p_t(f(\theta))$, the term used in SJC. The problem is that hidden in PF-ODE is the use of the (inverse) Euclidean metric to convert the differential-form score function into a vector field to update the parameters. In canonical coordinates, the Euclidean metric is the identity. As a result, the components of the score function do not change when they are transformed into a vector field by the Euclidean metric. Therefore, we can safely ignore the metric term in the original PF-ODE formulation for $\mathcal X$.
>
> If we want to convert the pulled-back score function into the corresponding pulled-back vector field, we need to use the pulled-back Euclidean metric inverse given by $(J^\top J)^{-1}$. When we use this metric, we get the same pulled-back form of the PF-ODE as given above.
>
> A more informal way of seeing why the correct update must look like the pulled-back ODE we have derived here rather than that given by SJC is to look at the case where the number of input and output dimensions is the same. From the chain rule, we have $\frac{d\theta}{dt} = \frac{d\theta}{dx} \frac{dx}{dt}$.
> We can compute $\frac{d\theta}{dx}$ using the inverse function theorem $\frac{d\theta}{dx} = \left(\frac{dx}{d\theta}\right)^{-1} = J^{-1}$, where $J$ is the Jacobian of $f$. So we arrive at $\frac{d\theta}{dt}=J^{-1}\frac{dx}{dt}$. When $f$ is invertible, this equation is equivalent to the one given above. For non-invertible $f$, we can interpret the pullback as the solution to the least squares minimization problem $\min\left\|J\frac{d\theta}{dt} -\frac{dx}{dt}\right\|^2$ for $\frac{d\theta}{dt}$.
>
> We thank you for your time spent reviewing. We have invested substantial effort in addressing your concerns and improving the quality of the paper, and we ask that you consider adjusting your score in light of our response. Please let us know if you have additional questions we can help answer.

---

### Official Review · Reviewer_BA4f · 2024-07-16

**Soundness:** 2
**Presentation:** 3
**Contribution:** 2
**Rating:** 5
**Confidence:** 1

**Summary:**

This paper introduces a novel, training-free method to sample through differentiable functions using pretrained diffusion models.

**Strengths:**

It sounds a general method that could apply to many different scenarios.

**Weaknesses:**

More systematic evaluation of the method in addition to image examples would be great.

**Questions:**

I do not have questions at this stage.

**Limitations:**

The authors have discusses some limitations.

---

> ### Author Rebuttal · Authors · 2024-08-07
>
> Thank you for your positive review. Please see the common response for additional SIREN panoramas. Let us know if there are any remaining concerns we might be able to address.

---

### Official Review · Reviewer_H3if · 2024-07-16

**Soundness:** 2
**Presentation:** 2
**Contribution:** 2
**Rating:** 5
**Confidence:** 3

**Summary:**

The present paper introduces a training-free method to sample differentiable representations using pre-trained diffusion models. This is achieved by pulling back the dynamics of the reverse-time process from the image to the parameter space. Moreover, training-free methods for conditional sampling are employed to (approximately) satisfy implicit constraints. Numerical experiments show the performance of the method for images, panoramas, and 3D NeRFs. In particular, it is shown that the method can improve upon the baselines in terms of quality and diversity of generated representations.

**Strengths:**

The proposed framework is versatile and can be used for various differentiable representations. The presented numerical experiments show that the resulting methods can improve quality and diversity compared to other training-free baselines.

**Weaknesses:**

1. Baselines and related works: Further discussion of related works and additional numerical comparisons are needed.
    - It would be helpful to provide further discussion on related works, such as Zero-1-to-3, HiFA, Magic123, LatentNeRF, Fantasia3D, ...
    - While the presented results are promising, additional experiments and visualizations would strengthen the validation of the method.
    - A more detailed comparison to methods requiring fine-tuning/training could provide a clearer picture of the method's performance. In particular, it would be good to add comparisons in terms of (overall) runtime/NFEs.
    - Why are there no PSNR/SSIM/LPIPS numbers in Table 1 for the baselines, e.g., SJC?
    - While DreamFusion is similar to SJC, it would still be interesting to add empirical comparisons.

2. Presentation and theory: Further details and explanations could be provided (e.g., also in the appendix).
    - The geometric perspective and pullback operation might be challenging for readers not well-versed in these areas. One could improve accessibility by providing more intuitive explanations.
    - Precise details on how RePaint is adapted to this setting seem to be missing. In this context it would also be good to clarify the connections to other methods for posterior sampling (DPS/ReSample/...).
    - It would be helpful to add some explanation on the "separately managed noise object" $\epsilon(t)$ or $\epsilon(\pi)$.

**Questions:**

Why is the score not the gradient of a scalar function?

Typos:
1. The visualizations in Fig. 1 (right) seem not to match the explanation "The example images in Figure 1 (right) show that the samples produced by our method are almost indistinguishable from the reference."
2. The order (top/bottom) in the caption of Fig. 4 is wrong.

**Limitations:**

Only a few limitations are mentioned and it would be good to enumerate potential failure cases of the proposed method.

---

> ### Author Rebuttal · Authors · 2024-08-07
>
> Thank you for your comments and review. We will address your concerns in turn.
>
> We have added a discussion of additional 3D asset generation from image diffusion methods like Zero-1-to-3, HiFA, Magic123, LatentNeRF, and Fantasia3D, which you mentioned. These works build off the Dreamfusion (SDS) / SJC method by adding additional inputs, finetuning, or regularization to improve the generation quality or by expanding the inputs for generating 3D assets. Zero-1-to-3 builds off of SJC using the same form of the score chaining, but with a score function fine-tuned to make use of a single input real image and additional view information. Magic 123 uses Zero-1-to-3 with some additional priors based on the 3D information. Fantasia3D separates geometry modeling and appearance into separate components, using SDS to update both. HiFA adds a different schedule and applies regularization to improve SDS. Finally, LatentNERF also uses SDS but parametrizes the object directly in the latent space of the stable diffusion autoencoder, rendered not to RGB values but instead into the dimensions of the latent space. These methods use the SDS/SJC backbone method for sampling/mode finding. In contrast, our work is focused on providing a more faithful sampling procedure to replace SDS/SJC for 3D generation and broader differentiable function sampling.
>
> Similarity metrics for SJC: Initially, we did not report PSNR, SSIM, and LPIPS for SJC because the generations have little relation to the ones generated by the vanilla diffusion model with the same seed. We have added these in the table below. Notice that they are far, far worse than our method (a better comparison is a measure of distributional similarity like KID, which we report in Figure 1). This is because SJC does not follow the same trajectory as the PF-ODE, so even if it were performing sampling, the samples would not look identical to those produced in the image space. The PSNR, SSIM, and LPIPS scores are used to compare images with identical content but with different compression levels; they are not designed to compare distributions of images.
>
> Table for metrics on SJC
> |  CFG  |          0 |         3 |        10 |        30 |       100 |
> |:------|-----------:|----------:|----------:|----------:|----------:|
> | psnr  | 8.50652    | 8.04771   | 7.13961   | 6.14309   | 5.54287   |
> | ssim  | 0.00938183 | 0.0419226 | 0.0988217 | 0.0973837 | 0.0843486 |
> | lpips | 1.10491    | 0.982072  | 0.855826  | 0.813387  | 0.787548  |
>
> Benchmarking Dreamfusion: SJC and Dreamfusion (SDS) are extremely similar in methodology when accounting for the conversion between the denoising model and the score function. The key differences are in the weighting function that the expectation is taken over, the optimization and sampled times in the schedule, and regularizers added for the 3D case. The SJC paper includes a section discussing the differences between the methods and provides a qualitative assessment of the assets generated by both methods.
>
> Precise details of how RePaint is applied: The most straightforward approach to encourage the pullback of the PF-ODE to produce intermediate samples that are representable by $f$ is to adapt the RePaint method (designed for inpainting) to this consistency constraint. RePaint takes advantage of the complete Langevin SDE diffusion process instead of the PF-ODE we have been considering here. It works by intermixing several forward and reverse steps in the schedule. Since RePaint requires a stochastic process to apply the conditioning, we use the DDIM \citep{song_denoising_2022} sampling procedure with $\eta=0.75$ for both the forward and backward steps.
>
> $\epsilon(t)$ and $\epsilon(\pi)$: Following the pulled-back PF-ODE we can find $\theta_t$ so that $f(\theta_t)$ represents a sample $x(t)\sim p_{0t}(x(t)|x(0)) = \mathcal N(x(t);x(0),\sigma^2(t)I)$. One limitation of this approach is that $x(t)$ is noisy, and differentiable image or 3D representations have a hard time expressing noise. In these settings, $J$ is ill-conditioned, leading to poor performance and long convergence times.
>
> To address this issue, we can factor $x(t)$ into the noiseless signal and noise using the reparameterization of the perturbation kernel $x(t) = \hat x_0(t) + \sigma(t)\epsilon(t)$. If we let $\epsilon(t) = \epsilon$ be constant throughout the sampling trajectory and start with $\hat x_0(T) = 0$, we can update $\hat x_0$ using $\frac{d\hat x_0}{dt} = \frac{dx}{dt} - \dot\sigma(t)\epsilon$. Using this decomposition, we let $f(\theta_t)$ represent the noiseless $\hat x_0(t)$ instead of $x(t)$, which makes it much more likely that $J$ is much better conditioned.
>
> Question: Why is the score not the gradient of a scalar function?
> Here, we use scalar in the differential geometry sense of being a scalar quantity on the given manifold, not an object that has a value dependent on the chosen coordinate chart. For continuous probability distributions, the probability density function varies not just on the chosen point but also on the given coordinate chart, as made evident by its changing value under a change of variables (where the additional $\det J$ term arises). Informally, this can be seen from the fact that $\int_R p_X(x)dx = \int_R p_Y(y)dy$, but because of the changing volume element $p_X\ne p_Y$. In contrast, a scalar function will have the same value when evaluated at the same point but in two distinct coordinate charts.
>
> We also agree that the method section could also be explained better. We have provided a revised method section in our response to Reviewer wR2k. We hope this more intuitive description of our method improves your consideration of our paper.
>
> We thank you for your time spent reviewing. We have invested substantial effort in addressing your concerns and improving the quality of the paper, and we ask that you consider adjusting your score in light of our response. Please let us know if you have additional questions we can help answer.

---

> > ### Comment · Reviewer_H3if · 2024-08-12
> >
> > I thank the authors for their rebuttal and explanations and raised my score. I understand the difference between mode finding and sampling, however, I still think that this should be further validated experimentally, e.g., further visualizations like Fig. 2 (also for 3D NeRFs) as well as comparison to other baselines (even though they are *conceptually* similar to SDS/SJC). In that spirit, I thank the authors for providing additional metrics but agree that other metrics should be considered that can compare distributions instead of instances.

---

### Author Rebuttal · Authors · 2024-08-07

Firstly, we would like to thank all the reviewers for their thoughtful comments and feedback on our submission. We appreciate the opportunity to address your concerns and clarify aspects of our work.

We want to reiterate the contribution of our work as a superior method to perform true sampling of differentiable representations (diffreps) using a pretrained diffusion model instead of merely mode finding as performed by DreamFusion and SJC. This allows our method to generate diverse and high-quality diffreps for a given prompt, even at low CFG levels. Also, we want to highlight that our method is training-free compared to methods like ProlificDreamer. This allows it to be applied directly with any score estimator regardless of architecture, making our contribution considerably more versatile and applicable to various domains and modalities.

We want to make an important note that may have given the impression that our method is not performing as well as it actually does in practice. In Figure 2, we have compared the raw image samples generated by following the PF-ODE in image space to the SIREN renders generated with our method and SJC. This figure is a substantively more faithful comparison of the generation abilities of the methods than Figure 1 (right). The results in Figure 1 (right) are presented out of order, giving the impression that there is no clear winner between the two methods and introducing the apparent tension with the statement, "samples produced by our method are almost indistinguishable from the reference." We apologize for this oversight and have corrected the figure, attaching it to the material below. The new Figure 1: (Corrected) demonstrates the clear improvement of our method, particularly at low CFG levels when the distinction between mode and typical samples is most relevant.

We provide a more detailed explanation of why DreamFusion and SJC are practically identical and why they perform mode-finding instead of sampling next.

In DreamFusion, they perform gradient ascent using $\mathbb{E}\_{t,\epsilon}[w(t)J^\top(\hat\epsilon(f(\theta) + \sigma(t)\epsilon)-\epsilon)],$ derived from the denoising objective, where $J$ is the Jacobian of the differentiable render function $f$ and $\hat{\epsilon}$ is the noise predictor. Score Jacobian Chaining (SJC) performs gradient ascent using $\nabla \log p(\theta):= \mathbb{E}_{t,\epsilon}[J^\top \nabla\log p_t(f(\theta)+\sigma(t)\epsilon)].$ If we rewrite the DreamFusion objective $\mathbb{E}\_{t,\epsilon}\big[w(t)J^\top (\hat{\epsilon}(f(\theta)+\sigma(t)\epsilon)-\epsilon)\big]$ using the Tweedie formula, we get $\mathbb{E}\_{t,\epsilon}\left[\tfrac{w(t)}{\sigma(t)}J^\top \nabla \log p_t(f(\theta)+\sigma(t)\epsilon)\right].$

This expression is identical to the $\nabla \log p(\theta)$ term from SJC if we let $w(t) = \sigma(t)$ for all $t$.

Both approaches estimate the objective using Monte Carlo sampling. In addition, SJC also uses a custom sampling schedule for the $t$s, which can be interpreted as gradient annealing to align the implicit $\sigma(t)$ of the diffrep with the $t$ used to evaluate the score function. Following the gradients to convergence leads to a critical point in the $\log p(f(\theta))$ landscape, finding a (local) maximum or mode. Neither procedure has a set stopping time, and running the PF-ODE using the parameter score function suggested by these methods would fail to produce samples from the distribution (consider the procedure on Gaussian data as a concrete example). Furthermore, while both methods can produce passable diffreps at high classifier-free guidance (CFG) levels, they struggle to produce coherent diffreps when the CFG weight is low, especially if the distribution is multimodal or has more degrees of freedom.

To appreciate the limitations of using the mode as a proxy for sampling, we need to consider two scenarios: the multi-modal setting and the high-dimensional setting. In the multi-modal setting, the sample distribution may contain several distinct high-density regions in the multi-modal setting. However, mode-finding algorithms typically focus on only one of these regions, potentially sacrificing sample diversity. This lack of diversity can be seen in the SJC samples (last row) of Figure 2.

Although the mode might look like a typical sample in low dimensions, the mode is anomalous when the sample space is high dimensional $d \gg 1$ (e.g., the space of images). This fact can be intuitively understood as a consequence of the thin-shell phenomenon, which states that while low-dimensional standard Gaussian samples concentrate around their mode, high-dimensional Gaussian samples predominantly reside in an exponentially thin shell around the boundary of a ball centered on the mode with a radius $\sqrt{d}$. As an illustrative example, consider sampling a normalized pure-noise image. Despite having a mode of $0$, we would almost never expect to generate a uniformly gray image. This provides some insight as to why the mode of a high-dimensional distribution will lack the quality and details present only in the samples from the thin shell.

We have also added more panorama images in the supplementary PDF.

Once more, we would like to highlight the uniqueness of our contribution as the true pullback of the diffusion process to the diffrep parameter space. Unlike fine-tuning methods like ProlificDreamer or other training-free methods, our approach provides a means to efficiently apply the dynamics of the diffusion process directly in the parameter space through the pullback to sample high-quality diffreps.

We thank the reviewers for their valuable insights and hope our responses have addressed their concerns. We believe that our work significantly contributes to the distillation of diffusion sampling space, and we hope this response alleviates any concerns the reviewers have.

---

### Decision · Program_Chairs · 2024-09-25

**Decision:**

Accept (poster)

**Comment:**

This paper proposes a zero-shot method for sampling implicit representations from pretrained diffusion models, which could in principle allow for better coverage of the full distribution, whereas prior methods might be more susceptible to mode collapse. The reviewers generally found the paper interesting and the contribution promising, but identified limited scope of experiments (for example the need for comparison to more baselines, and to validate the diversity claims in further settings) as a weakness, as well as some issues with clarity or errors in presentation. After discussion, some of these concerns were addressed — particularly presentational ones — but the concerns about the coverage of the experiments remain. Overall, the conceptual contribution seems potentially valuable and impactful, but it would need a more convincing and comprehensive demonstration of the benefits to achieve optimal impact.